# Different Uptake of ^68^Ga-FAPI and ^18^F-FDG in Lymphadenopathy Caused by Angioimmunoblastic T-Cell Lymphoma in a Patient with Colon Cancer

**DOI:** 10.3390/diagnostics12092211

**Published:** 2022-09-13

**Authors:** Meiqi Wu, Qingqing Pan, Yaping Luo

**Affiliations:** 1Department of Nuclear Medicine, Chinese Academy of Medical Sciences and Peking Union Medical College Hospital, Beijing 100730, China; 2Beijing Key Laboratory of Molecular Targeted Diagnosis and Therapy in Nuclear Medicine, Beijing 100730, China

**Keywords:** lymphadenopathy, lymphoma, ^18^F-FDG, ^68^Ga-FAPI, PET/CT

## Abstract

An 82-year-old man with a history of colon cancer was found with multiple lymphadenopathies and a pulmonary mass. Fluorine-18 fluorodeoxyglucose positron emission tomography/computed tomography (^18^F-FDG PET/CT) detected generalized hypermetabolic lymph nodes and an FDG-avid pulmonary mass. PET/CT with ^68^Ga-labeled fibroblast activation protein inhibitor (FAPI) revealed intense uptake in the lung mass, consistent with metastatic disease from colon cancer. However, the lymphadenopathies were not avid for ^68^Ga-FAPI, suggesting a different etiology. The biopsy of a cervical node confirmed angioimmunoblastic T-cell lymphoma. The case showcased the potential of ^68^Ga-FAPI in differentiation of solid tumor and hematological disease due to different histopathologic nature of stromal fibrosis.

Different uptake of ^68^Ga-FAPI and ^18^F-FDG in lymphadenopathy caused by angioimmunoblastic T-cell lymphoma in a patient with colon cancer.

**Figure 1 diagnostics-12-02211-f001:**
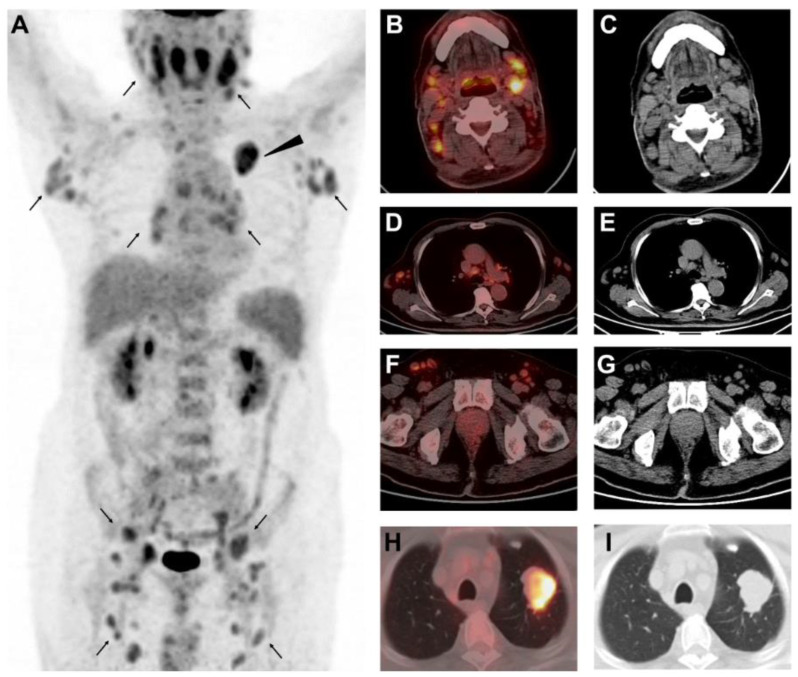
An 82-year-old man presented with cervical lymphadenopathy and low-grade fever for 3 weeks. He had a history of colon cancer (pT3N1bM0) and underwent right hemicolectomy followed by three cycles of chemotherapy 3 years ago. Laboratory examination showed elevated serum carcinoembryonic antigen (CEA, 24.9 ng/mL, normal range, 0–5.0), and contrast-enhanced CT detected a pulmonary mass and enlarged lymph nodes in the chest and abdomen. Fluorine-18 fluorodeoxyglucose positron emission tomography/ computed tomography (^18^F-FDG PET/CT) was performed for suspicion of metastasis from colon cancer. The MIP of PET (**A**) and axial fusion and CT images (**B**–**G**) showed multiple FDG-avid nodes symmetrically distributed in the cervical, mediastinal, axillary, retroperitoneal, pelvic, and inguinal regions (**A**–**G**, arrows, SUVmax 7.6). Additionally, an FDG-avid mass was noted in the left upper lobe of the lung (**A**. arrowhead, SUVmax 7.2; (**H**,**I**), axial fusion and CT image, lung window). Based on his history and serum CEA level, pulmonary metastasis from colon cancer was first suspected. The lymphadenopathy was generalized, hypermetabolic and symmetrically distributed, though voiding colic or mesentery lymph nodes. Such presentation was not characteristic of lymph node metastases from colon cancer; instead, they were more likely to be related to hematological disorder, such as lymphoma (adapted from [1]) or multi-center Castleman disease (adapted from [2]), or systemic inflammatory disease, such as virus infection [3], typhoid fever (adapted from [4]), sarcoidosis (adapted from [5]), necrotizing lymphadenitis [6], IgG4-related disease [7], and adult-onset Still’s disease (adapted from [8]).

**Figure 2 diagnostics-12-02211-f002:**
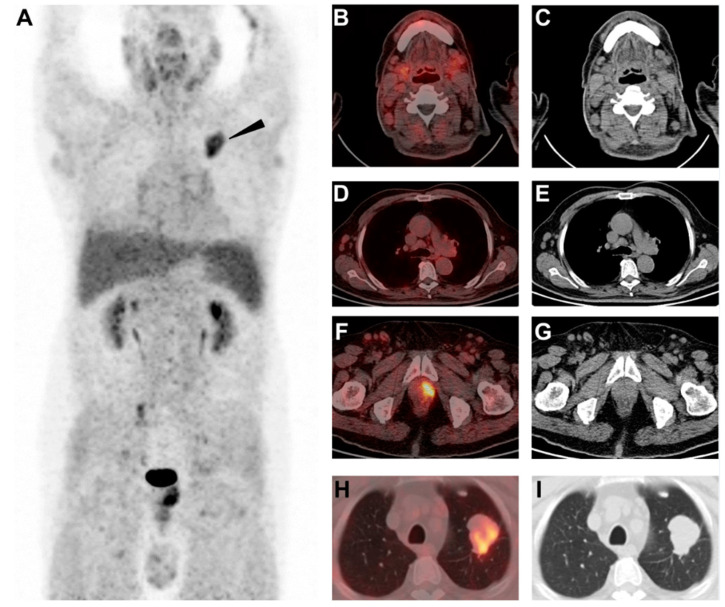
For further differentiation, he was included in a clinical trial of PET/CT with ^68^Ga-labeled fibroblast activation protein inhibitor (FAPI) approved by the institutional review board in our hospital (zs-1810), and written informed consent was obtained from the patient. ^68^Ga-FAPI is a recently introduced agent targeting cancer-associated fibroblasts (FAP) that overexpress in the stroma of various solid tumors [9,10,11]. However, hematological disease and certain types of inflammatory nodes (e.g., IgG4-related lymphadenopathy, rheumatoid disease) usually lack fibrosis (adapted from [12,13,14,15]). Based on this hypothesis, we thought ^68^Ga-FAPI might help differentiate metastatic lesions and other etiologies in this patient. ^68^Ga-FAPI PET/CT was performed three days after ^18^F-FDG PET/CT. The MIP of ^68^Ga-FAPI PET (**A**) and axial fusion and CT images (**B**–**G**) showed the previous hypermetabolic nodes were not avid for ^68^Ga-FAPI (SUVmax 1.6), but the mass in the upper left lung showed intense ^68^Ga-FAPI uptake ((**A**). arrowhead, SUVmax 6.4; (**H**,**I**), axial fusion and CT image, lung window). In addition, a FAPI-avid prostatic lesion was noted (**F**–**G**, SUVmax 7.9). As expected, we consider the pulmonary mass avid for both FDG and ^68^Ga-FAPI was metastatic from colon cancer, and the lymph nodes (FDG+/FAPI-) were possibly a hematological disorder, such as lymphoma. The accidentally detected lesion in the prostate (FDG-/FAPI+) might be prostate cancer.

The patient subsequently underwent excisional biopsy of a cervical node, and histopathological examination confirmed angioimmunoblastic T-cell lymphoma. He did not have a further lung biopsy due to complicated disease status. Combining the patient’s history, the elevated serum CEA level, and findings on PET/CTs (the smooth margin, singularity, and high uptake of both ^18^F-FDG and ^68^Ga-FAPI) of the lung lesion, metastasis from colon cancer or a primary lung cancer were primarily considered. He exhibited elevated serum prostate-specific antigen (PSA) level (21.9 ng/mL, normal range, 0–4.0). Prostate MRI detected abnormal signal in the prostate apex (central zone), consistent with the lesion shown on ^68^Ga-FAPI PET/CT, with decreased T2 signal and elevated DWI (PI-RADS 4). Thus, prostate cancer was highly suspected. The patient died 3 months later due to multiple organ failure before initiation of systemic therapy. ^18^F-FDG and ^68^Ga-FAPI are not tumor-specific agents, although enhanced glycolysis is the major characteristic of malignant tumors [16]. Although a few studies detected ^68^Ga-FAPI uptake in hematological malignancies [17,18], we think the different histopathologic nature of stromal fibrosis in solid tumor and hematological disease may still bring some potential of ^68^Ga-FAPI in differentiating these diseases, as in the current case.

## Data Availability

No new data were created or analyzed in this study. Data sharing is not applicable to this article.

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
