# Peer review of "Different Uptake of 68Ga-FAPI and 18F-FDG in Lymphadenopathy Caused by Angioimmunoblastic T-Cell Lymphoma in a Patient with Colon Cancer"

_diagnostics, 2022, doi:10.3390/diagnostics12092211_

Round 1

Reviewer 1 Report

The paper under evaluation (Interesting Image) shows the differential uptake of dual tracer PET/CT with 18F-FDG and 68Ga-FAPI in a patient previously treated for colorecal cancer, presenting lung mass coupled with generalized lymph adenopathies. 

While lung mass incorporated both 18F-FDG and 68Ga-FAPI, lymph adenopathies were positive only at 18F-FDG PET/CT, then resuling positive for angioimmunoblastic T cell lymphoma. 

The paper is well written and documented and represents an example of combined use of 18F-FDG and 68Ga-FAPI.

Here some suggestions:

- the interval between the 2 scans (18F-FDG and 68Ga-FAPI PET/CT) should be added to the main text;

- the authors state that institutional review board approved the FAPI PET, the number of the approved study should be added;

- As far as it concerns the prostate lesion, no biopsy was carried out, therefore, the findings should be interpreted with caution. At least, a brief comment on correlative MRI findings (including Pi-RADS) might be of value.

The main limitation of the paper is the lack of histological confirmation of lung metastasis; this should be more clearly stressed in the manuscript.

Author Response

We sincerely appreciate the reviewer’ constructive comments, which helped us to improve our manuscript. Please find out our point-by-point responses to these comments below.

Point 1: the interval between the 2 scans (18F-FDG and 68Ga-FAPI PET/CT) should be added to the main text;

Response 1: Thank you for the comments. 68Ga-FAPI PET/CT for the patient was performed three days after 18F-FDG PET/CT (Page 3, line 55).  

Point 2: the authors state that institutional review board approved the FAPI PET, the number of the approved study should be added;

Response 2: Thank you for the comment. The approval code (zs-1810) has been added in the figure legend and the IRB statement (Page 3, line 50; page 4, line 92).

Point 3: As far as it concerns the prostate lesion, no biopsy was carried out, therefore, the findings should be interpreted with caution. At least, a brief comment on correlative MRI findings (including Pi-RADS) might be of value.

Response 3: Thank you for the comment. A summary of the prostate MRI findings including PI-RADS has been added. "Prostate MRI detected abnormal signal in the prostate apex (central zone), consistent with the lesion shown on 68Ga-FAPI PET/CT, with decreased T2 signal and elevated DWI (PI-RADS 4)" (Page 3, line 69-71).

Point 4: The main limitation of the paper is the lack of histological confirmation of lung metastasis; this should be more clearly stressed in the manuscript.

Response 4: Thank you for the comment. We elaborated on the main clinical considerations of the lung lesion. "He did not have a further lung biopsy due to complicated disease status. Combining the patient’s history, the elevated serum CEA level, and findings on PET/CTs (the smooth margin, singularity, and high uptake of both 18F-FDG and 68Ga-FAPI) of the lung lesion, metastasis from colon cancer or primary lung cancer were primarily considered" (Page 3, line 65-68).

We truly appreciate your pertinent and constructive advice. We have benefitted from your concern and suggestions. Hopefully, our manuscript has been improved after revision according to your comments.

Reviewer 2 Report

This is an interesting images paper. I have no detailed comments. However, the authors can discuss and cite the recommended reference about tumor metabolism, especially the 18F-2-DG.

Tumor energy metabolism and potential of 3-bromopyruvate as an inhibitor of aerobic glycolysis: implications in tumor treatment. Cancers, 2019, 11, 317.

Author Response

Thank you for the comment. We added the reference as ref. 16 in the resubmission. "18F-FDG and 68Ga-FAPI are not tumor specific agents, though enhanced glycolysis is the major characteristic of malignant tumors[16]" (Page 3, line 72-73; page 4, line 132-133).